

# A pan-sharpening network using multi-resolution transformer and two-stage feature fusion

Wensheng Fan, Fan Liu and Jingzhi Li

College of Data Science, Taiyuan University of Technology, Jinzhong, Shanxi, China

## ABSTRACT

Pan-sharpening is a fundamental and crucial task in the remote sensing image processing field, which generates a high-resolution multi-spectral image by fusing a low-resolution multi-spectral image and a high-resolution panchromatic image. Recently, deep learning techniques have shown competitive results in pan-sharpening. However, diverse features in the multi-spectral and panchromatic images are not fully extracted and exploited in existing deep learning methods, which leads to information loss in the pan-sharpening process. To solve this problem, a novel pan-sharpening method based on multi-resolution transformer and two-stage feature fusion is proposed in this article. Specifically, a transformer-based multi-resolution feature extractor is designed to extract diverse image features. Then, to fully exploit features with different content and characteristics, a two-stage feature fusion strategy is adopted. In the first stage, a multi-resolution fusion module is proposed to fuse multi-spectral and panchromatic features at each scale. In the second stage, a shallow-deep fusion module is proposed to fuse shallow and deep features for detail generation. Experiments over QuickBird and WorldView-3 datasets demonstrate that the proposed method outperforms current state-of-the-art approaches visually and quantitatively with fewer parameters. Moreover, the ablation study and feature map analysis also prove the effectiveness of the transformer-based multi-resolution feature extractor and the two-stage fusion scheme.

## INTRODUCTION

Multi-spectral (MS) images are widely used in remote sensing applications such as land cover classification (*Ghamisi et al., 2019*), environmental change detection (*Bovolo et al., 2010*) and agriculture monitoring (*Gilbertson, Kemp & Van Niekerk, 2017*). Due to physical constraints, there is a trade-off between spatial and spectral resolutions during satellite imaging. The satellite can only provide low-spatial-resolution (LR) MS images and corresponding high-spatial-resolution (HR) PANchromatic (PAN) images (*Zhang, 2004*; *Zhou, Liu & Wang, 2021*). However, many applications mentioned above require satellite imagery with high spatial and spectral resolutions. Pan-sharpening meets the demand by fusing the LRMS and PAN images to obtain an HRMS image.

Corresponding author
Fan Liu, liufan@tyut.edu.cn

Various methods are proposed for pan-sharpening. They can be separated into four main classes: component substitution (CS), multi-resolution analysis (MRA), variational optimization (VO) and deep learning (DL) (*Vivone et al., 2015*; *Vivone et al., 2021b*; *Vivone et al., 2021a*). The first three classes belong to traditional algorithms that appeared several decades ago. The DL-based methods have emerged recently and achieved promising results.

The CS algorithms transform the LRMS image to another domain to replace its spatial component with the corresponding PAN data. Widely known CS approaches include those utilizing intensity-hue-saturation (IHS) transform (*Carper, Lillesand & Kiefer, 1990*), principal component analysis (PCA) (*Kwarteng & Chavez, 1989*), the Gram–Schmidt (GS) transform (*Laben & Brower, 2000*), and band-dependent spatial detail (BDSD) (*Garzelli, Nencini & Capobianco, 2008*).

The MRA category consists of algorithms that adopt a multi-scale decomposition to extract spatial details from the PAN image and inject them into up-sampled MS bands (*Xiong et al., 2021*). Wavelet transform (WT) (*Kim et al., 2011*), discrete wavelet transform (DWT) (*Pradhan et al., 2006*) and generalized Laplacian pyramids with modulation transfer function (MTF-GLP) (*Aiazzi et al., 2003*) are well-known MRA methods.

The VO methods rely on defining and solving optimization problems. For instance, P+XS (*Ballester et al., 2006*) obtains a high-fidelity HRMS image *via* a variational optimization process with several reasonable hypotheses. SR-D (*Vicinanza et al., 2015*) uses sparse dictionary elements to represent the desired spatial details. The representation coefficients are obtained by solving a variational optimization problem. In the last few years, VO methods have also been combined with DL techniques to use the advantages of both classes fully (*Shen et al., 2019*; *Deng et al., 2021*).

The DL class typically leverages data-driven learning to get an optimized solution for pan-sharpening. *Huang et al. (2015)* launched the first attempt at DL-based pan-sharpening by utilizing a modified sparse de-noising auto-encoder scheme. Inspired by convolutional neural networks (CNN) based image super-resolution (*Dong et al., 2016*), *Masi et al. (2016)* proposed an efficient three-layer CNN called PNN. To fully extract spectral and spatial features, RSIFNN (*Shao & Cai, 2018*) uses a two-stream architecture to extract features from the LRMS and PAN images separately. PanNet (*Yang et al., 2017*) generates spatial details with a deep residual network in the high-pass filtering domain for better spatial preservation. Spectral information is also preserved by directly adding the up-sampled LRMS image to the details. Observing that the scale of features varies among different ground objects, *Yuan et al. (2018)* proposed a multi-scale and multi-depth CNN called MSDCNN. Many follow-up works exploit multi-scale feature extraction in pan-sharpening. For example, PSMD-Net (*Peng et al., 2021*) embeds multi-scale convolutional layers into dense blocks for pan-sharpening. *Zhang et al. (2019)* proposed a bidirectional pyramid network that reconstructs the HRMS image from coarse resolution to fine resolution. Recently, several transformer-based pan-sharpening methods emerged (*Meng et al., 2022*; *Zhou et al., 2022*). They utilize transformers to extract long-range image features but have not considered multi-scale information. DR-NET (*Su, Li & Hua, 2022*) introduces Swin Transformer (*Liu et al., 2021*) blocks into a UNet-like architecture for spatial information preservation. CPT-noRef (*Li, Guo & Li, 2022*) uses a pyramid transformer encoder to

supply global and multi-scale features. With long-range feature extraction capacity, these transformer-based methods have achieved promising results. However, the utilization of transformers also induces higher model complexity. To avoid this, in this article, we aim to design a lightweight network that can exploit diverse features. Transformers in the proposed network only need to extract a few distinct features.

In existing DL-based methods, diverse features with multi-scale, multi-depth and contextual information are not fully extracted. And in image reconstruction, the indiscriminate use of these diverse features also limits the fusion quality. To solve these problems, we propose a pan-sharpening approach based on multi-resolution transformer and two-stage feature fusion in this article. Two transformer-based multi-resolution feature extractors (MRFE) are applied separately to the LRMS and PAN images to extract diverse features fully. After feature extraction, a multi-resolution fusion module (MRFM) and a shallow-deep fusion module (SDFM) are proposed to exploit multi-scale and multi-depth features for spatial detail generation. Finally, the generated details are injected into the up-sampled LRMS image to obtain the pan-sharpened image. Extensive experiments over QuickBrid (QB) and WorldView-3 (WV3) datasets demonstrate that the proposed method outperforms state-of-the-art algorithms visually and quantitatively. The main contributions can be summarized as follows:

1. A two-branch transformer-based feature extractor is designed to facilitate information interaction between different resolutions and finally learn effective multi-scale feature representation of the LRMS and PAN images.
2. An MRFM is proposed to fuse LRMS and PAN features at each resolution, which is simple yet effective for the fusion of multi-resolution modality-specific features.
3. An SDFM is proposed to fuse shallow local and deep multi-scale features, which is essential for fully utilizing diverse features.

## MATERIALS & METHODS

### Datasets

Pan-sharpening is a technique that uses the PAN image to sharpen the LRMS image. It has a wide application in the remote sensing field. However, there are various imaging satellites. The PAN and LRMS data they captured have different characteristics, which challenges the generality of pan-sharpening methods. Thus, two datasets captured by QB and WV3 satellites are used in the experiments to evaluate the performance of the proposed method. Another challenge is the absence of real HRMS images. Since the ground-truth (GT) HRMS images are unavailable for the pan-sharpening task, we follow Wald's protocol (*Wald, Ranchin & Mangolini, 1997*) to spatially degrade the LRMS and PAN images by a factor of 4 (the spatial-resolution gap between the PAN and LRMS images). Then, the original full-resolution LRMS images can be regarded as references, *i.e.,* GT HRMS images. All the images are cropped into PAN patches with size of $128 \times 128$ and LRMS patches with size of $32 \times 32$ to generate datasets. As a result, the QB dataset has 11,216 patch pairs, and the WV3 dataset has 11160 patch pairs. The 11216 QB patch pairs are randomly divided into 8974/1121/1121 (80%/10%/10%) pairs for the training, validation, and testing

**Table 1   Detailed information about the two datasets.** GSD denotes the ground sample distance, which describes the spatial resolution of remote sensing imagery.

| Satellite | Image type | #Bands | GSD | Patch size | #Patch pairs (training/validation /testing) |
|---|---|---|---|---|---|
| QuickBird | PAN | 1 | 0.6 m | 128 × 128 | 8974/1121/1121 |
|  | MS | 4 | 2.4 m | 32 × 32 | |
| WorldView-3 | PAN | 1 | 0.3 m | 128 × 128 | 8928/1116/1116 |
|  | MS | 8 | 1.2 m | 32 × 32 | |

sets, respectively. Similarly, the 11160 WV3 patch pairs are divided into 8928/1116/1116 (80%/10%/10%) pairs for training, validating and testing. Additionally, to evaluate the performance of models on real-world full-resolution data, the original LRMS and PAN images without spatial degradation are cropped into 1121 QB patch pairs and 1116 WV3 patch pairs for full-resolution testing sets. Details about the datasets are given in Table 1.

## Method
### Overall network architecture
The overall architecture of the proposed method is depicted in Fig. 1, where $\mathbf{X}_P \in \mathbb{R}^{H \times W \times 1}$, $\mathbf{X}_M \in \mathbb{R}^{\frac{H}{4} \times \frac{W}{4} \times B}$ and $\hat{\mathbf{Y}}_M \in \mathbb{R}^{H \times W \times B}$ represent the PAN image, the LRMS image and the fused image. $W$ and $H$ are the width and height of the PAN image. $B$ is the number of MS bands. First, shallow local features of up-sampled $\mathbf{X}_M$ and $\mathbf{X}_P$ are extracted by two convolution layers with kernel size of $3 \times 3$, respectively. These convolution layers are also referred to as pre-conv layers. Then, the shallow local features are fed into an MRFE to further extract deep multi-resolution feature maps of the two images. The feature maps at the same resolution are concatenated and fused by an MRFM. Finally, the deep and shallow features are merged by an SDFM and added to the up-sampled $\mathbf{X}_M$ to obtain the pan-sharpened image $\hat{\mathbf{Y}}_M$. The entire pan-sharpening process can be described as follows:

$$\mathbf{M}_S = \text{Pre-conv}(\uparrow \mathbf{X}_M) \tag{1}$$

$$\mathbf{P}_S = \text{Pre-conv}(\mathbf{X}_P) \tag{2}$$

$$\mathbf{M}_{HR}, \mathbf{M}_{MR}, \mathbf{M}_{LR}, \mathbf{P}_{HR}, \mathbf{P}_{MR}, \mathbf{P}_{LR} = \text{MRFE}(\mathbf{M}_S, \mathbf{P}_S) \tag{3}$$

$$\mathbf{F}_D = \text{MRFM}([\mathbf{M}_{HR}, \mathbf{P}_{HR}], [\mathbf{M}_{MR}, \mathbf{P}_{MR}], [\mathbf{M}_{LR}, \mathbf{P}_{LR}]) \tag{4}$$

$$\mathbf{Y}_D = \text{SDFM}([\mathbf{M}_S, \mathbf{P}_S, \mathbf{F}_D]) \tag{5}$$

$$\hat{\mathbf{Y}}_M = \mathbf{Y}_D + \uparrow \mathbf{X}_M \tag{6}$$

where $\uparrow \mathbf{X}_M$ represents the up-sampled LRMS image. $\mathbf{M}_S$ and $\mathbf{P}_S$ are shallow features of the LRMS and PAN images. $\mathbf{M}_{HR}$ and $\mathbf{P}_{HR}$ are HR feature maps. $\mathbf{M}_{MR}$ and $\mathbf{P}_{MR}$ are Middle-Resolution (MR) feature maps. $\mathbf{M}_{LR}$ and $\mathbf{P}_{LR}$ are LR feature maps. $\mathbf{F}_D$ denotes the deep features. $\mathbf{Y}_D$ is the generated spatial details injected into the up-sampled LRMS image. $[\cdot]$ represents concatenation at the channel dimension. The MRFE, MRFM and SDFM are key components of the proposed method, which will be elaborated in the following.

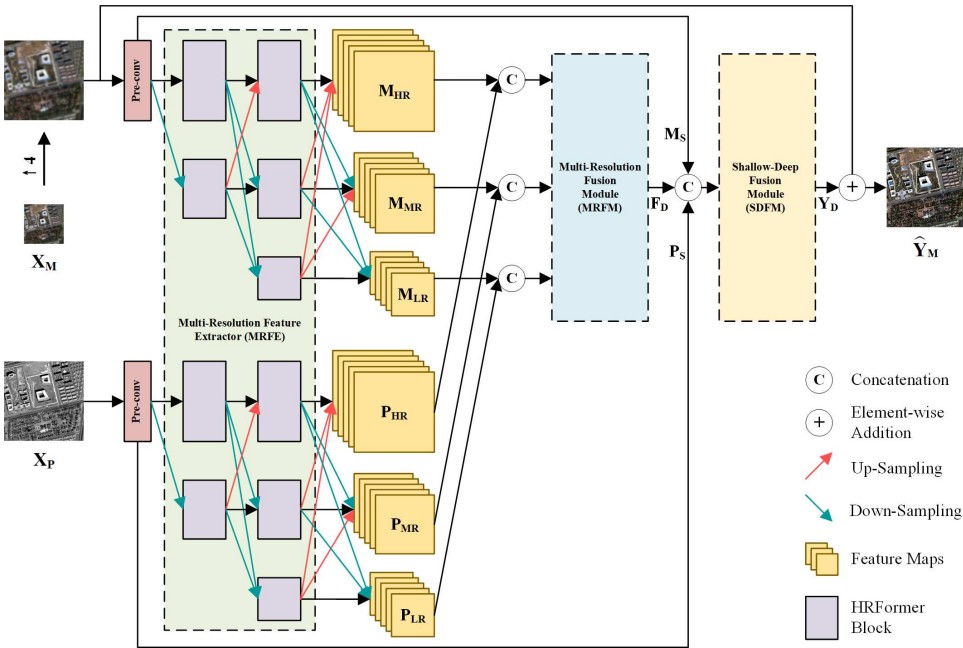

**Figure 1** **The overall architecture of the proposed pan-sharpening network.** ↑ 4 denotes up-sampling by a factor of 4 (the spatial resolution gap between $X_P$ and $X_M$). Pre-convolution (pre-conv) layers and the HRFormer-based MRFE are employed to extract diverse features. To fully exploit these features, a two-stage feature fusion strategy is adopted. The multi-resolution fusion is completed *via* the MRFM. The shallow-deep fusion is executed by the SDFM, which generates the spatial details $Y_D$. $Y_D$ is added to the up-sampled $X_M$ to obtain the pan-sharpened image. The photos in this figure are generated from the raw data available at https://github.com/zhysora/PSGan-Family.

## Multi-resolution feature extractor

Extracting effective and diverse features is of great importance to spatial information preservation. To this end, the MRFE keeps an HR stream without down-sampling to prevent spatial information loss and gradually adds MR and LR streams to extract multi-scale features. Furthermore, skip connections with up-sampling or down-sampling operations are used between multi-resolution streams to facilitate information interaction, which can help improve the features' effectiveness and reduce redundancy among streams. For long-range feature extraction, HRFormer blocks (*Yuan et al., 2021*) are used as basic blocks to build the MRFE. The structure of an HRFormer block is shown in Fig. 2. The Local-Window Self-Attention (LWSA) mechanism models long-range dependencies between pixels. Then, the Feed-Forward Network (FFN) with a 3 × 3 Depth-Wise (DW) convolution exchanges information across windows to acquire contextual information. As shown in Fig. 1, the MRFE consists of HRFormer blocks and skip connections between streams. Therefore, the HRFormer blocks can encode contextual information into the features. And the multi-resolution streams exchange information with each other to generate effective and diverse deep feature representations $M_{HR}$, $M_{MR}$, $M_{LR}$, $P_{HR}$, $P_{MR}$, and $P_{LR}$. In the following, two fusion stages are designed to fuse these features progressively.

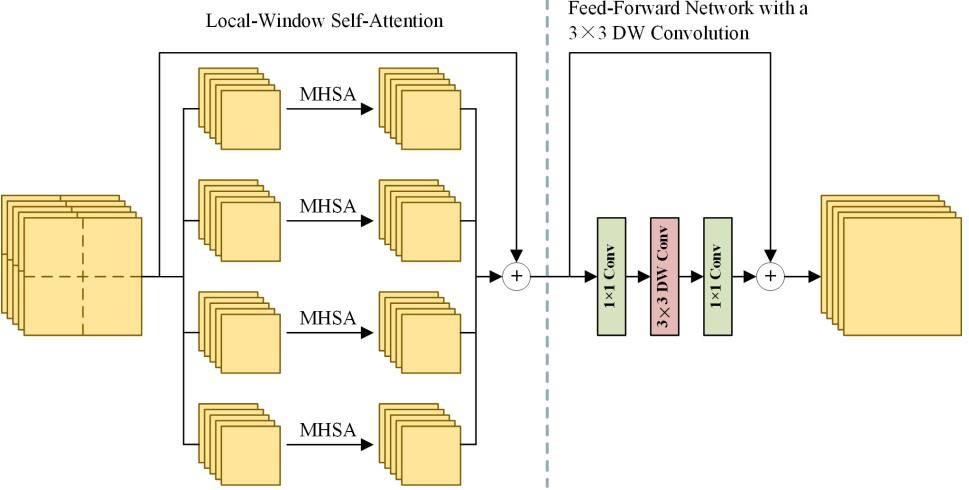

**Figure 2** **The structure of an HRFormer block.** MHSA denotes the multi-head self-attention mechanism. Left: the local-window self-attention splits feature maps into local windows and performs MHSA separately in each window. Right: the feed-forward network uses convolution layers to exchange information across local windows.

## Multi-resolution fusion module

In the first fusion stage, we propose an MRFM to fuse LRMS and PAN feature maps at each resolution. The structure of the MRFM is shown in Fig. 3. Every two LRMS and PAN feature representations with the same resolution are concatenated and fed into a $3 \times 3$ convolution layer to fuse the modality-specific features. Then, a residual block (ResBlock) (*He et al., 2016*) is used to refine the fused features. To restore the spatial details of the MR and LR feature maps, $3 \times 3$ convolution and pixel shuffle layers are used as learnable up-sampling procedures. Finally, feature maps with multi-scale information are concatenated to constitute the deep features $\mathbf{F}_D$. Thanks to the different depths and resolutions of these streams, the feature maps in $\mathbf{F}_D$ are diverse and complementary to the shallow features.

## Shallow-deep fusion module

To generate spatial details by fully using the shallow features $\mathbf{M}_S$, $\mathbf{P}_S$, and the deep features $\mathbf{F}_D$, an SDFM is proposed, which helps the network focus on more informative features and avoids the degradation problem (*He et al., 2016*). Figure 4 displays the structure of the SDFM. $\mathbf{M}_S$, $\mathbf{P}_S$ and $\mathbf{F}_D$ are concatenated by the channel dimension and fed into a $1 \times 1$ convolution to primarily fuse the features. Note that the shallow features $\mathbf{M}_S$ and $\mathbf{P}_S$ skip the MRFM. They are directly fed to this stage to preserve the original image information. Then, a standard squeeze and excitation block (*Hu, Shen & Sun, 2018*) is adopted to excite informative features. Specifically, a global average pooling (GAP) is used to aggregate the channel-wise global information into a channel descriptor. Subsequently, two $1 \times 1$ convolution layers reduce and restore the dimension of the descriptor to capture the correlations among channels. The restored descriptor is mapped to a set of channel weights *via* a sigmoid function. Thus, informative channels can be excited by scaling the primarily

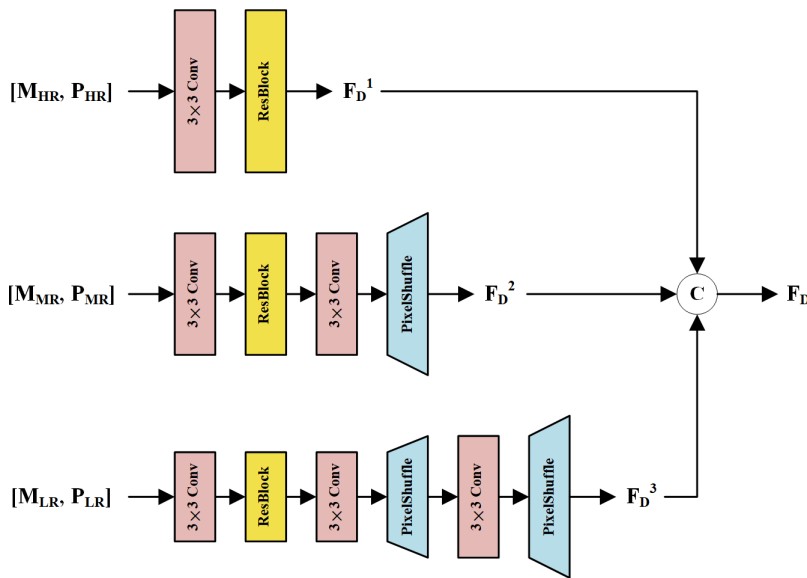

**Figure 3** **The structure of the MRFM.** The MS and PAN feature maps at the same resolution are concatenated and fused *via* a corresponding subnetwork. The subnetworks up-sample the feature maps *via* pixel shuffle layers, resulting in high-resolution feature maps $\mathbf{F}_D^1$, $\mathbf{F}_D^2$, and $\mathbf{F}_D^3$. These feature maps are concatenated to form the fused deep features $\mathbf{F}_D$.

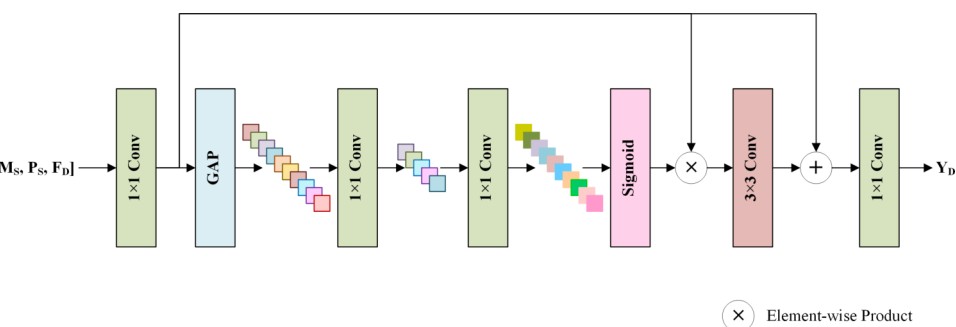

**Figure 4** **The structure of the SDFM.** The shallow features $\mathbf{M}_S$, $\mathbf{P}_S$, and deep features $\mathbf{F}_D$ are concatenated and fused *via* convolution layers and a standard channel attention mechanism. The output is the spatial details $\mathbf{Y}_D$, which will be added to the up-sampled LRMS image to obtain the pan-sharpened image.

fused features with the channel weights. We exploit the excited features to learn a residual component to refine the fused features. A 3 × 3 convolution layer completes this step. Finally, a 1 × 1 convolution generates spatial details $\mathbf{Y}_D$ from the refined features. The details are injected into the up-sampled LRMS image to obtain the pan-sharpening result $\hat{\mathbf{Y}}_M$.

# RESULTS

## Metrics

Five widely used indicators are adopted to evaluate different methods quantitatively. The indices can be grouped into four full-reference indicators and one no-reference indicator according to whether they require a GT image in calculations. For the evaluation on reduced-resolution datasets, we measure the four full-reference indices, including Spectral Angle Mapper (SAM) (*Yuhas, Goetz & Boardman, 1992*), relative dimensionless global error in synthesis (ERGAS) (*Wald, 2002*), spatial Correlation Coefficient (sCC) (*Zhou, Civco & Silander, 1998*), and the $Q2^n$ (*Alparone et al., 2004*; *Garzelli & Nencini, 2009*) index (*i.e.,* Q4 for 4-band data and Q8 for 8-band data). ERGAS and $Q2^n$ evaluate the global quality of pan-sharpened results. SAM estimates spectral distortions. sCC measures the quality of spatial details. In the evaluation on full-resolution datasets, we adopt the no-reference index Hybrid Quality with No Reference (HQNR) (*Aiazzi et al., 2014*) with its spectral distortion component $D_\lambda$ and spatial distortion component $D_S$ to measure the quality of pan-sharpened results.

## Experimental setting

The proposed method is implemented with the PyTorch framework and runs on an NVIDIA GEFORCE RTX 3090 GPU. Our model is trained for 500 epochs by an AdamW (*Loshchilov & Hutter, 2019*) optimizer with an initial learning rate of 0.0005, a momentum of 0.9, $\beta_1 = 0.9$, $\beta_2 = 0.999$, and a weight decay coefficient of 0.05. The mini-batch size is set to 16.

The hyper-parameter setting of the proposed method is listed in Table 2. Since the shallow pre-conv layers mainly focus on local regions and extract fine-grained features with rich spatial details, the channel number of output feature maps at these shallow layers is set to $C_S = 16$. The deep multi-scale feature maps in the MRFE and MRFM have contextual information. Contextual information is essential to pan-sharpening because of the similarity among ground objects. However, the pan-sharpening task focuses more on fine-grained spatial details than contextual semantic information. Therefore, the number of output feature maps at each layer in the MRFE and MRFM is set to $C_D = 8$. In the SDFM, except for the last layer, the number of feature maps equals the total feature map amount of $\mathbf{M}_S$, $\mathbf{P}_S$ and $\mathbf{F}_D$ (*i.e.,* 56). $K$ is the window size of the LWSA in an HRFormer block, which is set to 8 by default. $H$ denotes the head number of the MHSA in the LWSA. $H = 1$ is enough as the channel number $C = 8$ is quite low.

Experiments are conducted to verify the proposed model configuration. First, under the condition of a roughly unchanged total number of feature maps, the channel number of shallow features $C_S$ and the channel number of deep features $C_D$ are adjusted to find an appropriate setting of feature map numbers. The mean and standard deviation (STD) of the experimental results are reported in Table 3, which demonstrates that the fine-grained shallow feature maps at fine should be slightly more than the coarse-grained deep feature maps. We tested different values to determine the MHSA's head number $H$. The results are listed in Table 4, which proves that $H = 1$ is enough.

**Table 2** **The architecture configuration of the proposed method.** The PAN and MS feature extractors share the same configuration. Therefore, the settings of one pre-conv and MRFE branch are listed for brevity. Downsp. Rate denotes the downsampling rate of a network stream.

| Downsp. Rate | Pre-conv | MRFE | MRFM | SDFM |
|---|---|---|---|---|
| 1× | $3 \times 3, 16$ | $\begin{bmatrix} \text{LWSA}, K=8, H=1 \\ \text{FFN}, C=8 \end{bmatrix} \times 2$ | $3 \times 3, 8; \text{ResBlock}$ | $1 \times 1, 56 \begin{bmatrix} \text{GAP}; 1 \times 1, 14; 1 \times 1, 56 \\ \text{Sigmoid}; 3 \times 3, 56 \end{bmatrix} 1 \times 1, B$ |
| 2× | | $\begin{bmatrix} \text{LWSA}, K=8, H=1 \\ \text{FFN}, C=8 \end{bmatrix} \times 2$ | $3 \times 3, 8; \text{ResBlock} \begin{bmatrix} 3 \times 3, 32 \\ \text{Pixel Shuffle} \end{bmatrix}$ | |
| 3× | | $\begin{bmatrix} \text{LWSA}, K=8, H=1 \\ \text{FFN}, C=8 \end{bmatrix}$ | $3 \times 3, 8; \text{ResBlock} \begin{bmatrix} 3 \times 3, 32 \\ \text{Pixel Shuffle} \end{bmatrix} \times 2$ | |

**Table 3** **Quantitative comparison (mean ± STD) of model variants with different numbers of shallow and deep feature maps on the reduced-resolution QB testing set.** The best results are bold. $C_S$ is the number of shallow feature maps. CD is the number of deep feature maps.

| $C_S$ | $C_D$ | SAM | ERGAS | sCC | Q4 | $D_\lambda$ | $D_S$ | HQNR |
|---|---|---|---|---|---|---|---|---|
| 10 | 10 | $1.1602 \pm 0.4216$ | $0.8120 \pm 0.3043$ | $0.9908 \pm 0.0053$ | $0.9451 \pm 0.0674$ | $0.0307 \pm 0.0284$ | $0.0469 \pm 0.0473$ | $0.9249 \pm 0.0650$ |
| 16 | 8 | $\mathbf{1.1138 \pm 0.4043}$ | $\mathbf{0.7738 \pm 0.2898}$ | $\mathbf{0.9917 \pm 0.0049}$ | $\mathbf{0.9488 \pm 0.0651}$ | $\mathbf{0.0267 \pm 0.0211}$ | $\mathbf{0.0426 \pm 0.0392}$ | $\mathbf{0.9324 \pm 0.0527}$ |
| 20 | 6 | $1.1662 \pm 0.4237$ | $0.8147 \pm 0.3044$ | $0.9907 \pm 0.0053$ | $0.9451 \pm 0.0664$ | $0.0281 \pm 0.0288$ | $0.0444 \pm 0.0482$ | $0.9298 \pm 0.0651$ |
| Ideal value | | 0 | 0 | 1 | 1 | 0 | 0 | 1 |

**Table 4** **Quantitative comparison of model variants with different head numbers on the reducedresolution QB testing set.** The best results are bold. $H$ is the number of heads in the MHSA mechanism.

| $H$ | SAM | ERGAS | sCC | Q4 | $D_\lambda$ | $D_S$ | HQNR |
|---|---|---|---|---|---|---|---|
| 1 | $\mathbf{1.1138 \pm 0.4043}$ | $\mathbf{0.7738 \pm 0.2898}$ | $\mathbf{0.9917 \pm 0.0049}$ | $\mathbf{0.9488 \pm 0.0651}$ | $\mathbf{0.0267 \pm 0.0211}$ | $\mathbf{0.0426 \pm 0.0392}$ | $\mathbf{0.9324 \pm 0.0527}$ |
| 2 | $1.1575 \pm 0.4212$ | $0.8070 \pm 0.3019$ | $0.9909 \pm 0.0053$ | $0.9455 \pm 0.0671$ | $0.0304 \pm 0.0258$ | $0.0464 \pm 0.0447$ | $0.9255 \pm 0.0617$ |
| 4 | $1.1655 \pm 0.4227$ | $0.8138 \pm 0.3033$ | $0.9908 \pm 0.0053$ | $0.9449 \pm 0.0672$ | $0.0289 \pm 0.0240$ | $0.0448 \pm 0.0448$ | $0.9284 \pm 0.0604$ |
| Ideal value | 0 | 0 | 1 | 1 | 0 | 0 | 1 |

## Comparison with other methods

The proposed method is compared with eight widely used pan-sharpening techniques, including two CS algorithms: GSA (*Aiazzi, Baronti & Selva, 2007*), BDSD (*Garzelli, Nencini & Capobianco, 2008*), one MRA method: MTF-GLP-FS (*Vivone, Restaino & Chanussot, 2018*), one VO-based method: TV (*Palsson, Sveinsson & Ulfarsson, 2014*), two CNN-based methods: PNN (*Masi et al., 2016*), MSDCNN (*Yuan et al., 2018*), and two transformer-based methods: DR-NET (*Su, Li & Hua, 2022*) and *Zhou et al. (2022)*.

## Experimental results on reduced-resolution datasets

To verify the effectiveness of the proposed method, we conducted comparative experiments on the reduced-resolution QB and WV3 datasets. In this case, the original LRMS images are GT images for visual assessment.

Figure 5 displays visual results on the reduced-resolution QB data. To highlight the differences, we visualize residual maps between the pan-sharpening results and the reference (GT) in Fig. 6. A pixel with a small mean absolute error (MAE) is shown in blue. In contrast, a pixel with a big MAE is displayed in yellow. From the enlarged box

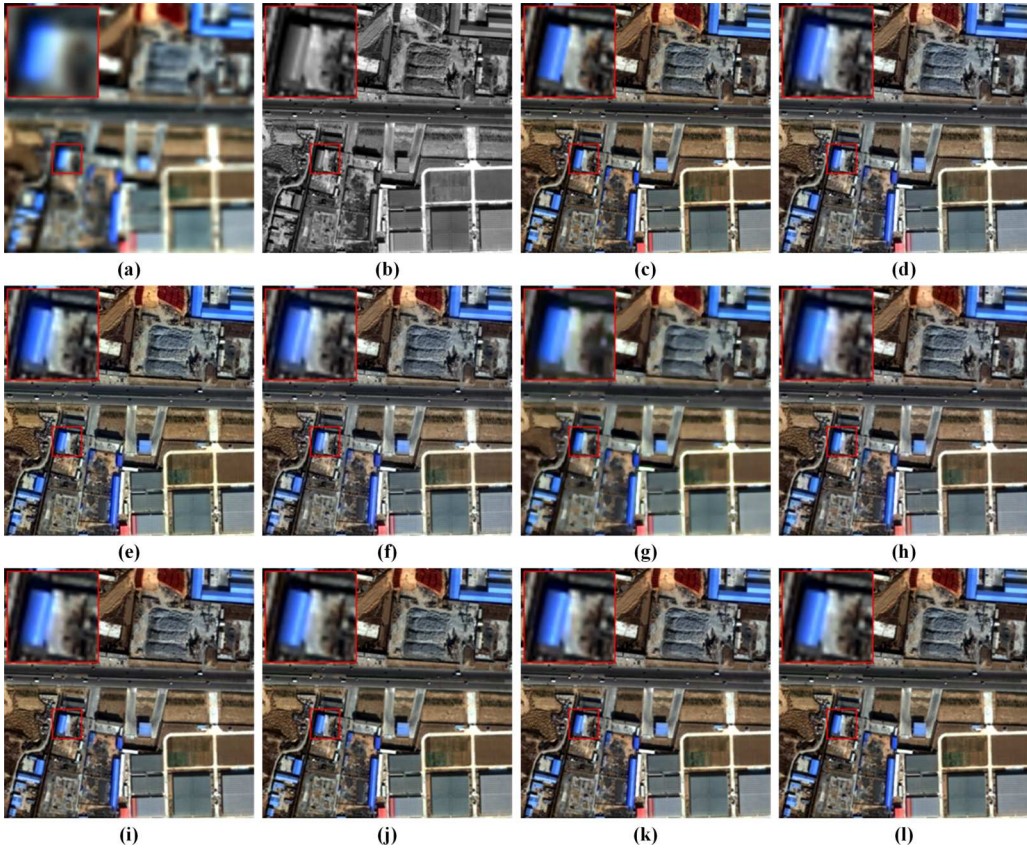

**Figure 5  Pan-sharpening results of different methods on a reduced-resolution QB testing patch pair.**
(A) Up-LRMS. (B) PAN. (C) GT. (D) GSA. (E) BDSD. (F) MTF-GLP-FS. (G) TV. (H) PNN. (I) MSD-CNN. (J) DR-NET. (K) *Zhou et al. (2022)* (L) Ours. Up-LRMS denotes the LRMS image up-sampled to the PAN size. The photos in this figure are generated from the raw data available at https://github.com/zhysora/PSGan-Family.

and the red buildings in the bottom center of images, it can be observed that results from GSA, MTF-GLP-FS and PNN have severe spectral distortions. The result of TV suffers from blurring effects. As shown in the enlarged views in Fig. 6, BDSD, MSDCNN, DR-NET and *Zhou et al. (2022)* show distinct yellow points, while the residual map of our MRPT is almost dark blue, which demonstrates the proposed method's superiority.

Fusion results on the reduced-resolution WV3 data are displayed in Fig. 7. Figure 8 displays the corresponding residual maps. As shown in the enlarged views of Fig. 7, the result of GSA, BDSD, MTF-GLP-FS and TV suffer from spectral distortions. From the enlarged views of residual maps in Fig. 8, it can be found that the results of PNN, MSDCNN, DR-NET and *Zhou et al. (2022)* have larger MAE in local areas than our method.

Tables 5 and 6 list the average performance and STD of different methods across all testing reduced-resolution image pairs. It can be found that transformer-based *Zhou et al. (2022)* and DR-NET have the second-best and third-best results on both datasets. Over the reduced-resolution QB testing set, our method yields the best quantitative results on SAM,

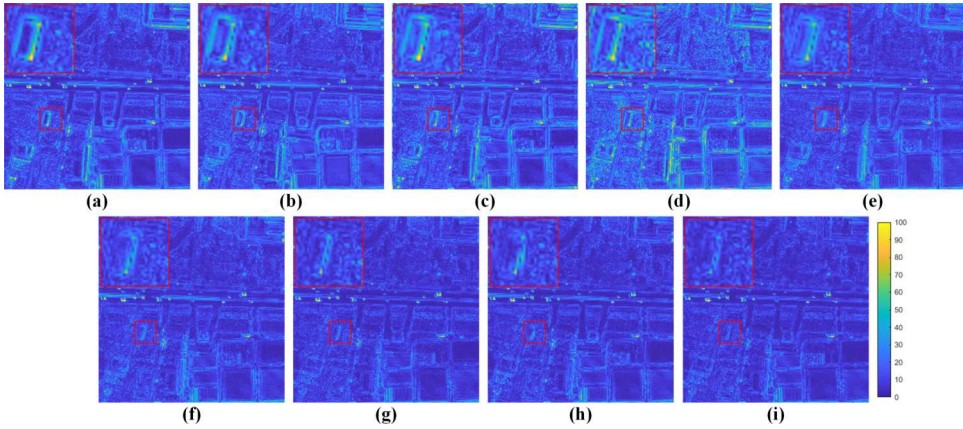

**Figure 6** Residual maps between the results and the GT in **Fig. 5**. (A) GSA. (B) BDSD. (C) MTF-GLP-FS. (D) TV. (E) PNN. (F) MSDCNN. (G) DR-NET. (H) *Zhou et al. (2022)* (I) Ours. The photos in this figure are generated from the raw data available at https://github.com/zhysora/PSGan-Family.

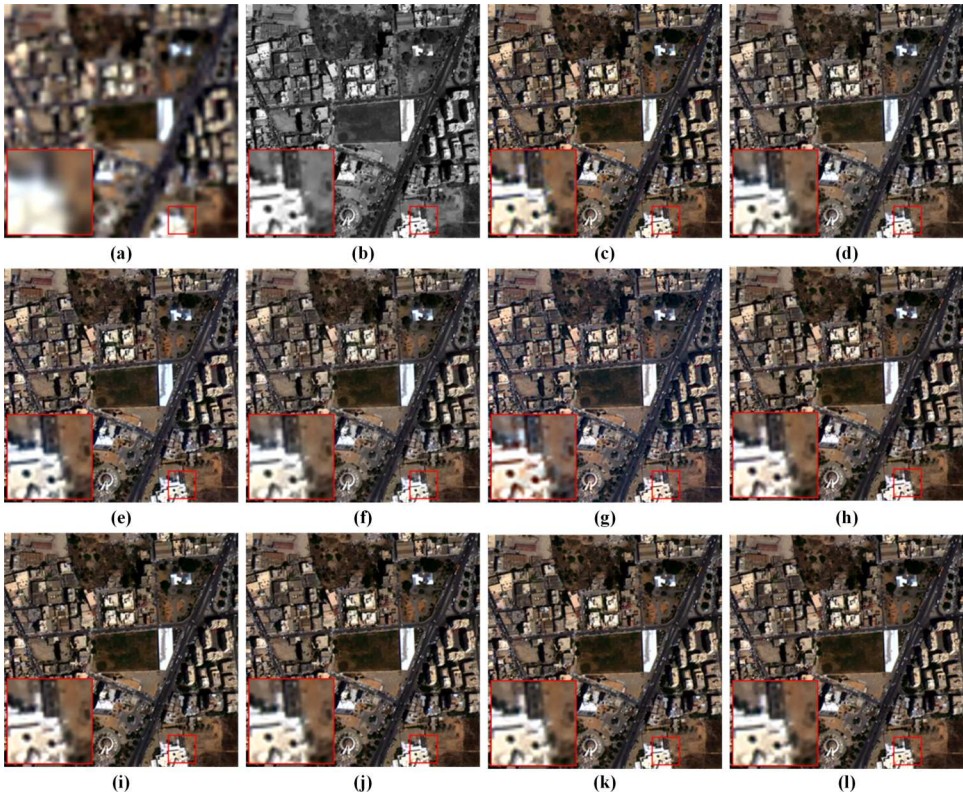

**Figure 7** Pan-sharpening results of different methods on a reduced-resolution WV3 testing patch pair. (A) Up-LRMS. (B) PAN. (C) GT. (D) GSA. (E) BDSD. (F) MTF-GLP-FS. (G) TV. (H) PNN. (I) MSDCNN. (J) DR-NET. (K) *Zhou et al. (2022)* (L) Ours. The photos in this figure are generated from the raw data available at https://github.com/zhysora/PGMAN.

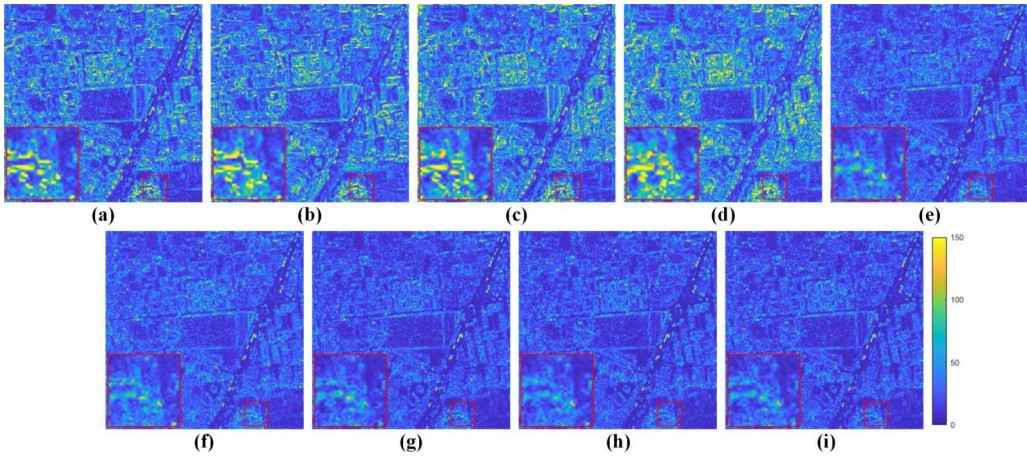

**Figure 8  Residual maps between the results and the GT in Fig. 7.** (A) GSA. (B) BDSD. (C) MTF-GLP-FS. (D) TV. (E) PNN. (F) MSDCNN. (G) DR-NET. (H) *Zhou et al. (2022)* (I) Ours. The photos in this figure are generated from the raw data available at https://github.com/zhysora/PGMAN.

**Table 5  Quantitative comparison of different methods on the reduced-resolution QB testing set.** The best results are bold and the second-best results are underlined.

|  | SAM | ERGAS | sCC | Q4 |
|---|---|---|---|---|
| GSA | $1.9273 \pm 0.8478$ | $1.4668 \pm 0.6854$ | $0.9678 \pm 0.0251$ | $0.8788 \pm 0.0931$ |
| BDSD | $1.9180 \pm 0.8243$ | $1.4297 \pm 0.6622$ | $0.9736 \pm 0.0193$ | $0.8888 \pm 0.0829$ |
| MTF-GLP-FS | $1.8771 \pm 0.8068$ | $1.4589 \pm 0.6426$ | $0.9691 \pm 0.0234$ | $0.8812 \pm 0.0871$ |
| TV | $1.9741 \pm 0.8194$ | $1.8726 \pm 0.9622$ | $0.9656 \pm 0.0191$ | $0.8245 \pm 0.1316$ |
| PNN | $1.4217 \pm 0.5216$ | $1.0570 \pm 0.3631$ | $0.9850 \pm 0.0096$ | $0.9107 \pm 0.0866$ |
| MSDCNN | $1.2629 \pm 0.4563$ | $0.9011 \pm 0.3213$ | $0.9886 \pm 0.0064$ | $0.9328 \pm 0.0747$ |
| DR-NET | $\underline{1.1280 \pm 0.4101}$ | $\underline{0.7855 \pm 0.2957}$ | $\mathbf{0.9918 \pm 0.0048}$ | $\underline{0.9475 \pm 0.0662}$ |
| Zhou et al. | $1.1601 \pm 0.4196$ | $0.8130 \pm 0.3062$ | $0.9907 \pm 0.0054$ | $0.9447 \pm 0.0675$ |
| Ours | $\mathbf{1.1138 \pm 0.4043}$ | $\mathbf{0.7738 \pm 0.2898}$ | $\underline{0.9917 \pm 0.0049}$ | $\mathbf{0.9488 \pm 0.0651}$ |
| Ideal value | 0 | 0 | 1 | 1 |

ERGAS and Q4. Furthermore, over the WV3 testing set, our method has the best results for all indicators.

## Experimental results on full-resolution datasets

All the methods are tested on original PAN and LRMS images to further verify the effectiveness of our method on the full-resolution real data. However, there are no GT HRMS images as references in this circumstance.

Figure 9 shows visual results on the full-resolution QB data. As shown in the enlarged view, the fusion result of GSA suffers from severe spectral distortions. It can also be observed that the results of BDSD, MTF-GLP-FS, PNN and DR-NET have slight color distortions. On the other hand, despite well-maintained spectral information, the results of TV, MSDCNN and *Zhou et al. (2022)* suffer from blurring effects. By comparison, our method presents the best pan-sharpening result regarding spectral and spatial fidelity.

**Table 6 Quantitative comparison of different methods on the reduced-resolution WV3 testing set.**
The best results are bold and the second-best results are underlined.

| | SAM | ERGAS | sCC | Q4 |
|---|---|---|---|---|
| GSA | 5.1082 ± 2.3376 | 3.8059 ± 1.7799 | 0.9455 ± 0.0395 | 0.8779 ± 0.1224 |
| BDSD | 5.6257 ± 2.6578 | 4.0366 ± 1.8778 | 0.9536 ± 0.0281 | 0.8688 ± 0.1314 |
| MTF-GLP-FS | 5.0723 ± 2.2696 | 3.8957 ± 1.7825 | 0.9466 ± 0.0365 | 0.8746 ± 0.1176 |
| TV | 5.2736 ± 2.0988 | 4.1111 ± 1.5417 | 0.9435 ± 0.0309 | 0.8538 ± 0.1356 |
| PNN | 3.4963 ± 1.2863 | 2.6023 ± 1.2673 | 0.9810 ± 0.0132 | 0.9126 ± 0.1228 |
| MSDCNN | 3.4077 ± 1.1944 | 2.4896 ± 1.1989 | 0.9829 ± 0.0123 | 0.9218 ± 0.1213 |
| DR-NET | 2.9762 ± 1.2064 | 2.2273 ± 1.2094 | 0.9871 ± 0.0119 | 0.9298 ± 0.1152 |
| *Zhou et al. (2022)* | 3.0180 ± 1.1999 | 2.2440 ± 1.1637 | 0.9861 ± 0.0114 | 0.9286 ± 0.1156 |
| Ours | **2.9452 ± 1.1927** | **2.1842 ± 1.1480** | **0.9871 ± 0.0112** | **0.9308 ± 0.1165** |
| Ideal value | 0 | 0 | 1 | 1 |

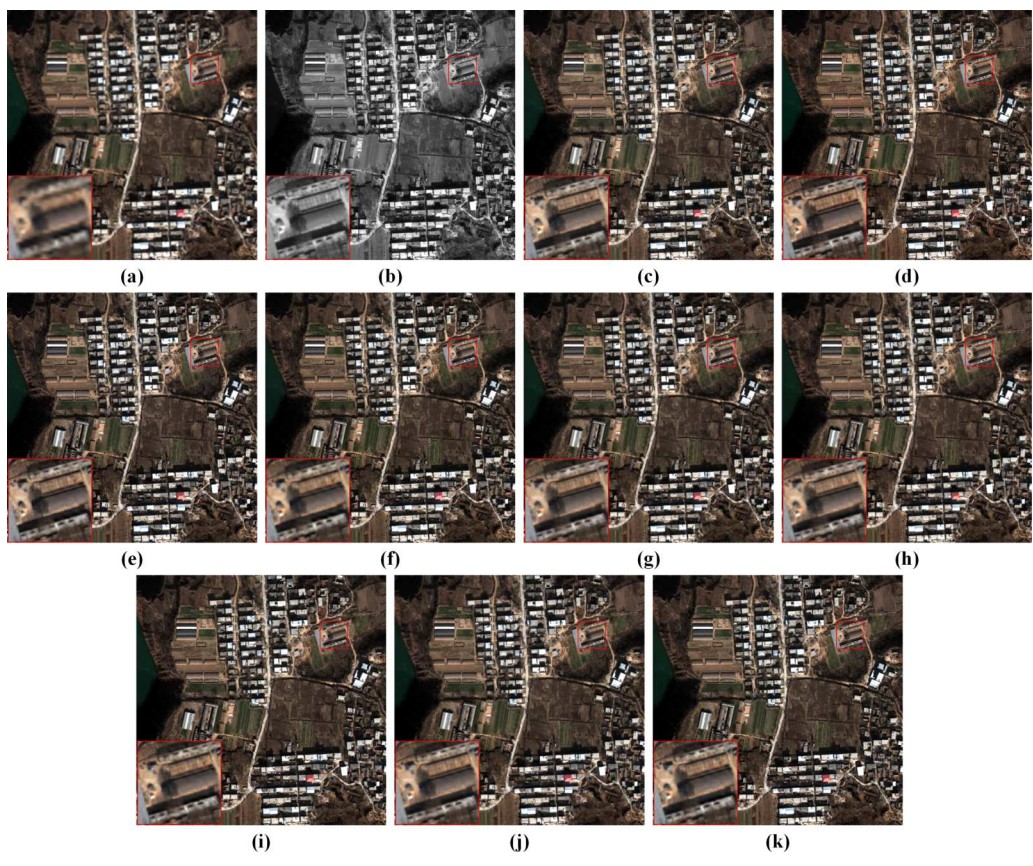

**Figure 9 Pan-sharpening results of different methods on a full-resolution QB testing patch pair.** (A) Up-LRMS. (B) PAN. (C) GSA. (D) BDSD. (E) MTF-GLP-FS. (F) TV. (G) PNN. (H) MSDCNN. (I) DR-NET. (J) *Zhou et al. (2022)* (K) Ours. The photos in this figure are generated from the raw data available at https://github.com/zhysora/PSGan-Family.

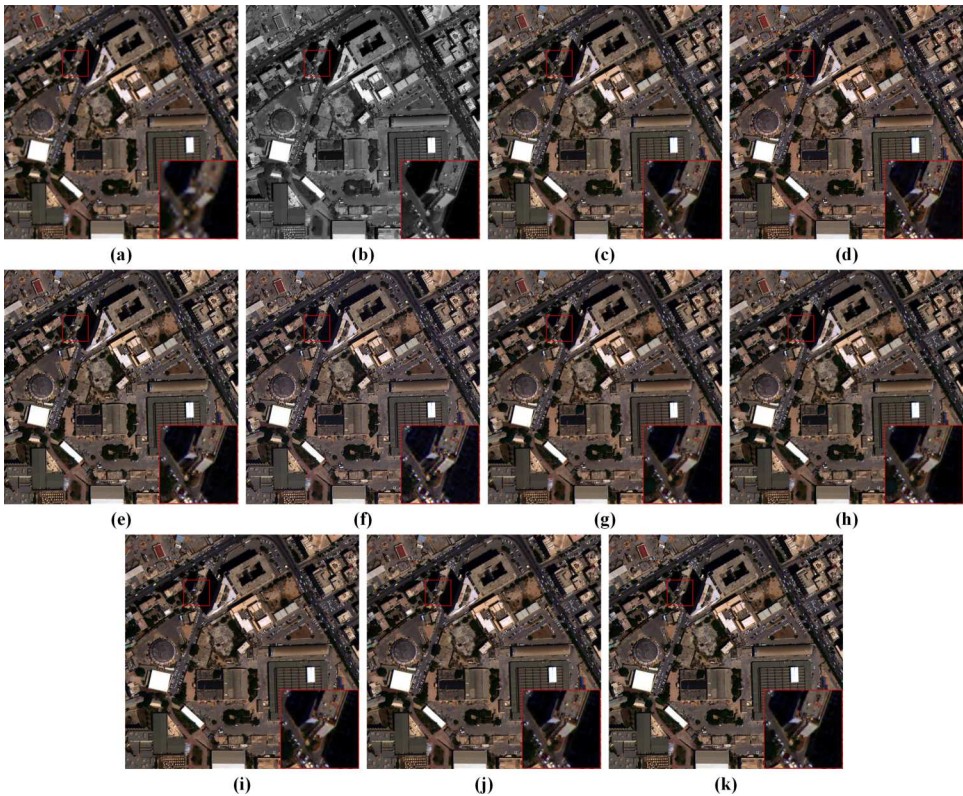

**Figure 10** **Pan-sharpening results of different methods on a full-resolution WV3 testing patch pair.**
(A) Up-LRMS. (B) PAN. (C) GSA. (D) BDSD. (E) MTF-GLP-FS. (F) TV. (G) PNN. (H) MSDCNN. (I)
DR-NET. (J) *Zhou et al. (2022)* (K) Ours. The photos in this figure are generated from the raw data available at https://github.com/zhysora/PGMAN.

Figure 10 displays visual results on the WV3 testing set at full resolution. From the enlarged views of Fig. 10, it can be observed that TV suffers from artifacts. The results of GSA, BDSD, MTF-GLP-FS, PNN, DR-NET and *Zhou et al. (2022)* have slight spectral distortions. MSDCNN suffers from blurring effects. Our method yields a fusion image with more explicit details and higher spectral fidelity.

The quantitative results of different methods on the two datasets are listed in Tables 7 and 8. Over the QB dataset, our method yields the best quantitative results on all three indicators. As for the WV3 testing set, TV has the best $D_\lambda$, and *Zhou et al. (2022)* has the best $D_S$. Our method has the second-best results on both indicators and the best HQNR value, which indicates that the proposed method has the best overall performance.

## Parameter numbers and time performance

To further evaluate the complexity of the proposed method, the parameter number and average runtime of each method on the 1,121 QB reduced-resolution testing patches are given in Table 9. The traditional algorithms are tested on a 2.6-GHz Intel Core i7-10750H CPU, and the DL-based approaches are tested on an NVIDIA GeForce RTX 2060 GPU. By comparison, the VO-based TV consumes more runtime than other methods. PNN

**Table 7  Quantitative comparison of different methods on the full-resolution QB testing set.** The best results are bold and the second-best results are underlined.

|  | $D_\lambda$ | $D_S$ | HQNR |
|---|---|---|---|
| GSA | $0.0796 \pm 0.0713$ | $0.1216 \pm 0.0980$ | $0.8146 \pm 0.1328$ |
| BDSD | $0.0701 \pm 0.0470$ | $0.0734 \pm 0.0647$ | $0.8636 \pm 0.0908$ |
| MTF-GLP-FS | $0.0413 \pm 0.0337$ | $0.0887 \pm 0.0786$ | $0.8748 \pm 0.0895$ |
| TV | $0.1398 \pm 0.1483$ | $0.0692 \pm 0.0390$ | $0.8024 \pm 0.1511$ |
| PNN | $0.0619 \pm 0.0700$ | $0.0554 \pm 0.0572$ | $0.8887 \pm 0.0986$ |
| MSDCNN | $0.0498 \pm 0.0582$ | $0.0514 \pm 0.0507$ | $0.9031 \pm 0.0862$ |
| DR-NET | $0.0498 \pm 0.0636$ | $0.0495 \pm 0.0467$ | $0.9050 \pm 0.0891$ |
| *Zhou et al. (2022)* | $0.0333 \pm 0.0290$ | $0.0486 \pm 0.0493$ | $0.9208 \pm 0.0675$ |
| Ours | $\mathbf{0.0267 \pm 0.0211}$ | $\mathbf{0.0426 \pm 0.0392}$ | $\mathbf{0.9324 \pm 0.0527}$ |
| Ideal value | 0 | 0 | 1 |

**Table 8  Quantitative comparison of different methods on the full-resolution WV3 testing set.** The best results are bold and the second-best results are underlined.

|  | $D_\lambda$ | $D_S$ | HQNR |
|---|---|---|---|
| GSA | $0.0829 \pm 0.0885$ | $0.0924 \pm 0.0929$ | $0.8392 \pm 0.1390$ |
| BDSD | $0.1446 \pm 0.1084$ | $0.0734 \pm 0.0639$ | $0.7969 \pm 0.1370$ |
| MTF-GLP-FS | $0.0484 \pm 0.0481$ | $0.0681 \pm 0.0751$ | $0.8889 \pm 0.0982$ |
| TV | $\mathbf{0.0395 \pm 0.0471}$ | $0.0771 \pm 0.0768$ | $0.8883 \pm 0.0980$ |
| PNN | $0.0819 \pm 0.0988$ | $0.0610 \pm 0.0897$ | $0.8696 \pm 0.1435$ |
| MSDCNN | $0.0798 \pm 0.1025$ | $0.0584 \pm 0.0870$ | $0.8737 \pm 0.1443$ |
| DR-NET | $0.0698 \pm 0.0654$ | $0.0571 \pm 0.0843$ | $0.8819 \pm 0.1219$ |
| *Zhou et al. (2022)* | $0.0558 \pm 0.0519$ | $\mathbf{0.0509 \pm 0.0784}$ | $0.8996 \pm 0.1081$ |
| Ours | $0.0467 \pm 0.0450$ | $0.0510 \pm 0.0809$ | $\mathbf{0.9078 \pm 0.1064}$ |
| Ideal value | 0 | 0 | 1 |

exhibits the lowest time consumption because it only has three convolution layers. Thus, its computational complexity is extremely low. Other methods show relatively close time consumption. In terms of parameter numbers, the parameters of DR-NET are considerably more than other methods. It is due to the large number of feature maps throughout the DR-NET. Exploiting these feature maps leads to redundancy and high complexity. The proposed method has the fewest parameters, which means low spatial complexity and will facilitate model deployment on devices with limited memory resources.

## Ablation study

Since the MRFE with skip connections, MRFM and SDFM are the core of our method, a series of ablation experiments are conducted to validate their effectiveness. Four variants of our network are built for the ablation study: (a) w/o MRFE; (b) w/o MRFM; (c) w/o SDFM; (d) w/o skip connections (w/o SC). The structures of these variants are shown in Fig. 11. The quantitative results of ablation experiments are listed in Table 10. Figure 12 shows the visual results of the variants over the QB dataset. Corresponding residual maps

**Table 9  Parameter number and average runtime comparison of different methods on the reduced-resolution QB testing set.** The runtimes' units are seconds (s). The parameters in a deep neural network may not have a straightforward physical interpretation. Thus, they have no units.

| Method | #Params | Runtime (s) |
| --- | --- | --- |
| GSA | – | 0.0067 |
| BDSD | – | 0.0121 |
| MTF-GLP-FS | – | 0.0211 |
| TV | – | 0.4026 |
| PNN | 80,420 | 0.0013 |
| MSDCNN | 189,852 | 0.0038 |
| DR-NET | 2,619,017 | 0.0120 |
| *Zhou et al. (2022)* | 70,600 | 0.0235 |
| Ours | 70,156 | 0.0170 |

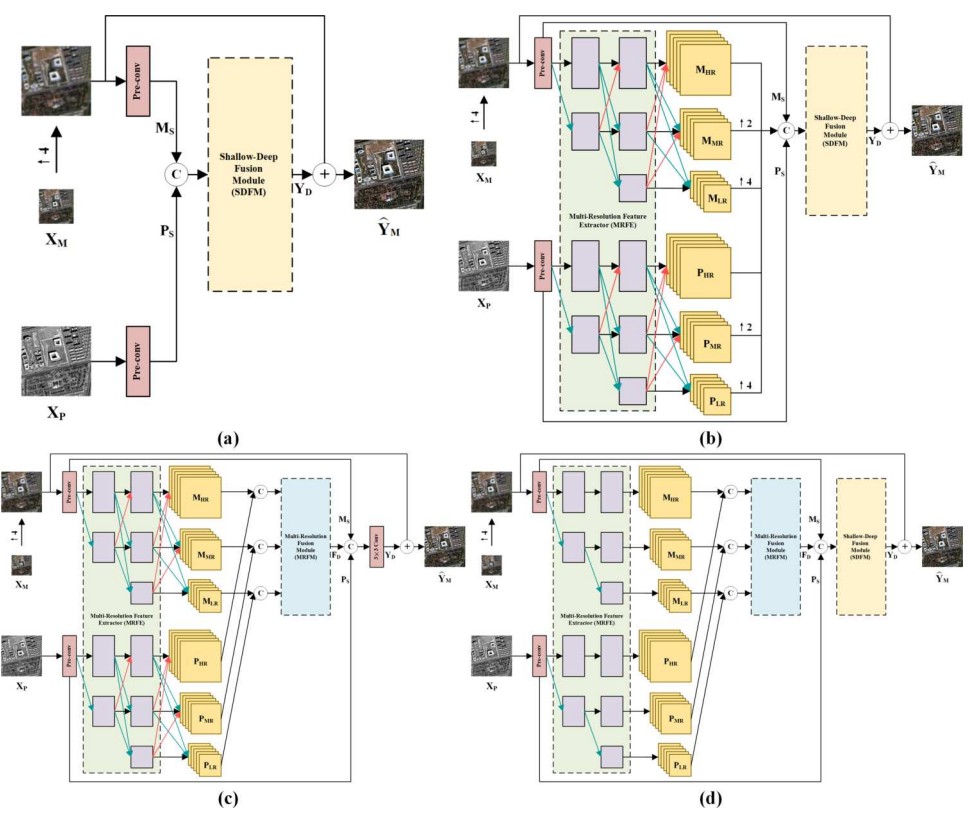

**Figure 11  Structure diagrams of the model variants in the ablation study.** (A) W/o MRFE. (B) w/o MRFM. (C) w/o SDFM. (D) w/o SC. The photos in this figure are generated from the raw data available at https://github.com/zhysora/PSGan-Family.

are displayed in Fig. 13. Figure 14 shows the convergence performance of the variants during the training process.

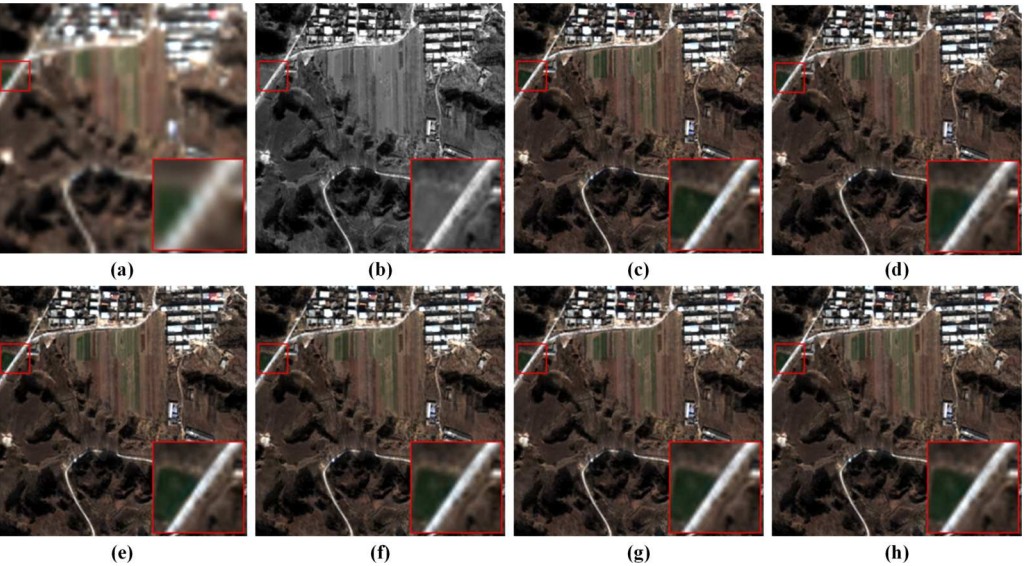

**Figure 12** **Pan-sharpening results of the ablation study on a reduced-resolution QB testing patch pair.**
(A) Up-LRMS. (B) PAN. (C) GT. (D) w/o MRFE. (E) w/o MRFM. (F) w/o SDFM. (G) w/o SC. (H) Ours.
The photos in this figure are generated from the raw data available at https://github.com/zhysora/PSGan-Family.

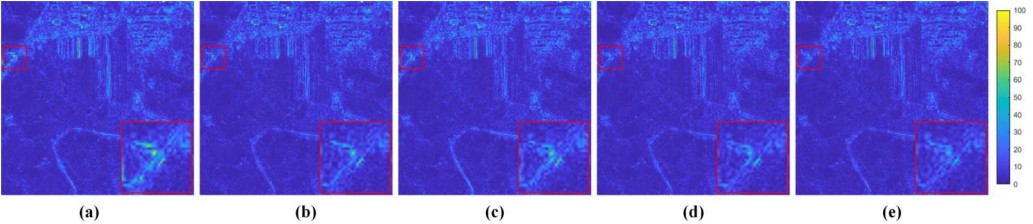

**Figure 13** **Residual maps between the results and the GT in Fig. 12.** (A) w/o MRFE. (B) w/o MRFM.
(C) w/o SDFM. (D) w/o SC. (E) Ours. The photos in this figure are generated from the raw data available
at https://github.com/zhysora/PSGan-Family.

## Importance of the multi-resolution feature extractor

The MRFE extracts deep multi-scale feature representations. The MRFM subsequently
fuses these representations. Therefore, removing the MRFE leads to the simultaneous
ablation of the MRFM. In the ablation study settings, the variant w/o MRFE eliminates
both the MRFE and MRFM, while the variant w/o MRFM only eliminates the MRFM. As a
result, the influence of the MRFE can be found by comparing the results of the two variants.
As shown in Table 10, the variant w/o MRFM yields much better results than the variant
w/o MRFE. From the residual maps in Fig. 13, it can also be observed that the variant w/o
MRFE shows larger residuals than the variant w/o MRFM. The comparison between these
two variants demonstrates that the MRFE plays a significant role in the proposed method.

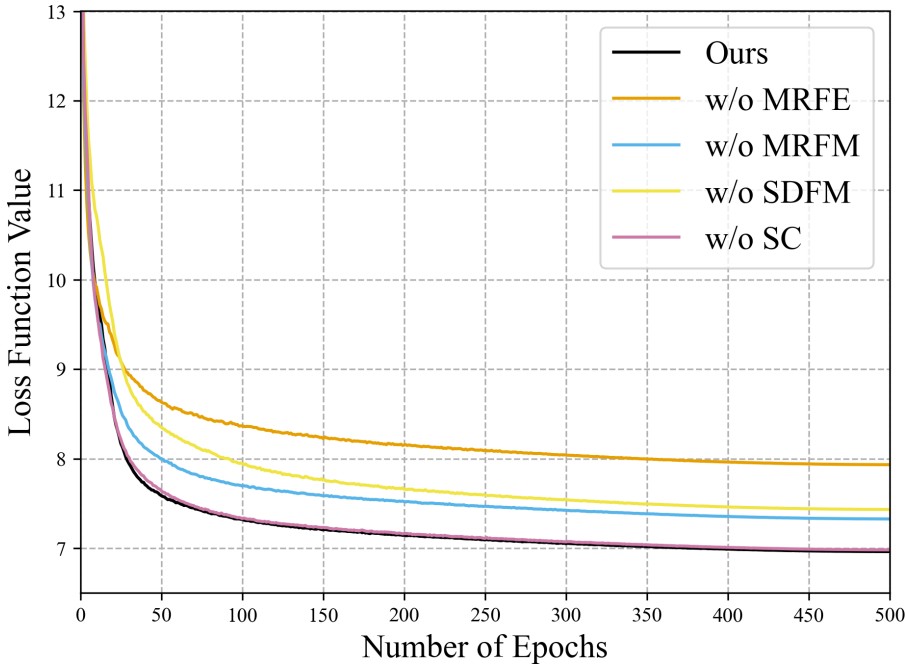

**Figure 14  Convergence performance of the variants in the training phase of the ablation study on the QB data set.** The loss function values of the first several epochs are much bigger than 13, which are omitted to highlight the final differences.

**Table 10  Quantitative comparison of the variants in the ablation study on the QB testing set.** The best results are bold.

| Variant | SAM | ERGAS | sCC | Q4 | $D_\lambda$ | $D_S$ | HQNR |
|---|---|---|---|---|---|---|---|
| w/o MRFE | 1.2678 ± 0.4643 | 0.8921 ± 0.3321 | 0.9888 ± 0.0066 | 0.9377 ± 0.0705 | 0.0504 ± 0.0474 | 0.0560 ± 0.0571 | 0.8987 ± 0.0879 |
| w/o MRFM | 1.1506 ± 0.4201 | 0.7997 ± 0.2999 | 0.9911 ± 0.0052 | 0.9465 ± 0.0665 | 0.0280 ± 0.0247 | 0.0455 ± 0.0455 | 0.9287 ± 0.0612 |
| w/o SDFM | 1.1824 ± 0.4352 | 0.8248 ± 0.3163 | 0.9903 ± 0.0057 | 0.9446 ± 0.0666 | **0.0264 ± 0.0205** | 0.0450 ± 0.0376 | 0.9303 ± 0.0498 |
| w/o SC | 1.1182 ± 0.4064 | 0.7750 ± 0.2894 | **0.9917 ± 0.0049** | 0.9484 ± 0.0656 | 0.0277 ± 0.0228 | 0.0435 ± 0.0422 | 0.9307 ± 0.0567 |
| Ours | **1.1138 ± 0.4043** | **0.7738 ± 0.2898** | **0.9917 ± 0.0049** | **0.9488 ± 0.0651** | 0.0267 ± 0.0211 | **0.0426 ± 0.0392** | **0.9324 ± 0.0527** |
| Ideal value | 0 | 0 | 1 | 1 | 0 | 0 | 1 |

## Importance of the multi-resolution fusion module

The MRFM is removed as the variant w/o MRFM in Fig. 11 to verify the necessity of two-stage feature fusion. In the variant w/o MRFM, both multi-resolution and shallow-deep features are fused by the SDFM. The results of the variant w/o MRFM in Table 10 are apparently inferior to our full model, which demonstrates that the MRFM is effective. These results also prove the two-stage feature fusion is better than the traditional one-shot fusion. Furthermore, the residual map in Fig. 13 also shows that removing the MRFM is harmful.

## Importance of the shallow-deep fusion module

To verify the effectiveness of the SDFM, we replace the module with a simple 3 × 3 convolution layer. The metrics of the variant w/o SDFM in Table 10 show that replacing

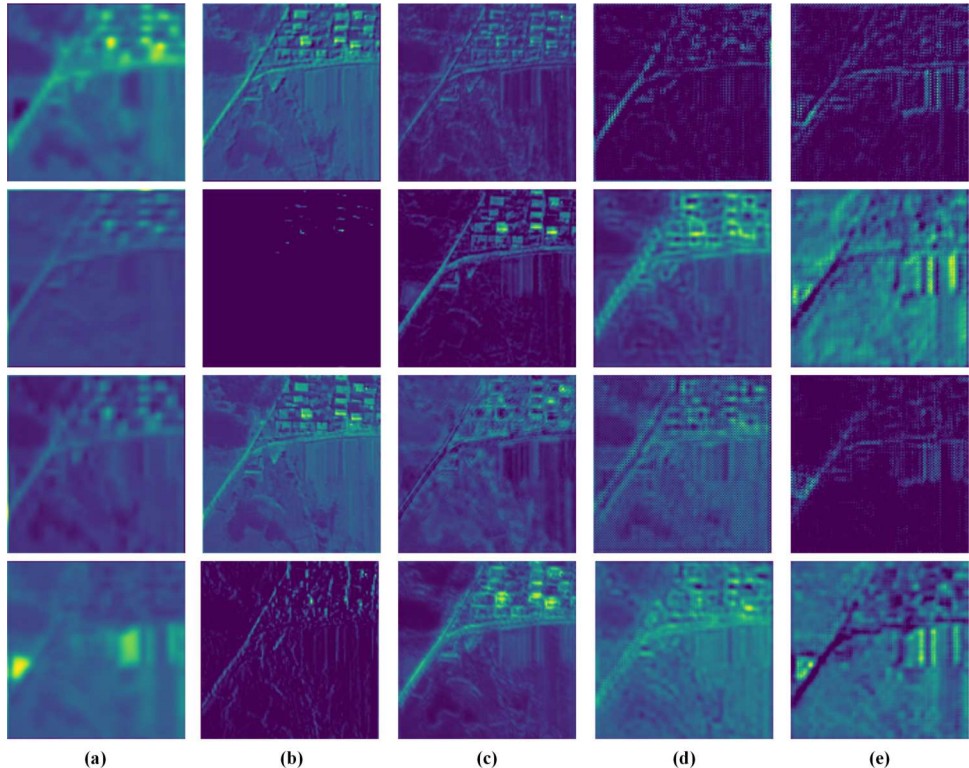

**Figure 15** **Visualization of the feature maps from the Pre-conv layers and the MRFM.** (A) $\mathbf{M}_S$. (B) $\mathbf{P}_S$. (C) $\mathbf{F}_D{}^1$. (D) $\mathbf{F}_D{}^2$. (E) $\mathbf{F}_D{}^3$. The photos in this figure are generated from the raw data available at https://github.com/zhysora/PSGan-Family.

the SDFM with a simple convolution layer is detrimental to fusion results. In Fig. 14, it is evident that the variant w/o SDFM converges slower than other structures, especially in the first 50 epochs. The slow convergence of the variant w/o SDFM demonstrates that a slightly more complicated shallow-deep fusion module can boost the convergence speed of our method.

## Importance of the skip connections between resolutions

Information exchange across resolutions relies on up-sampling and down-sampling skip connections. All the connections except those adding the streams are deleted in the variant w/o SC to investigate their influence on feature extraction. The results of the variant w/o SC in Table 10 show that all metrics are slightly down, which demonstrates that the skip connections are helpful.

## Visualization and analysis of feature maps

To verify the proposed network's diverse feature extraction ability, we visualize the feature maps extracted from a QB testing image pair in Fig. 15. These feature maps are finally fused by the SDFM and relate directly to the quality of pan-sharpening results.

The $\mathbf{M}_S$ and $\mathbf{P}_S$ feature maps are shallow features of the MS and PAN images. It can be observed that the $\mathbf{M}_S$ feature maps are blurry and lack spatial details. The $\mathbf{P}_S$ feature maps

are clear and some are high-frequency detail features. For instance, the $\mathbf{P}_S$ feature maps at the second and fourth rows in Fig. 15 have rich high-frequency details.

The $\mathbf{F}_D^1$, $\mathbf{F}_D^2$ and $\mathbf{F}_D^3$ feature maps are multi-scale deep features obtained from the MRFM. The feature maps from different resolutions have distinct content. The $\mathbf{F}_D^1$ are HR features and thus have rich spatial details. The MR features $\mathbf{F}_D^2$ contributes both detailed and contextual information. In LR features $\mathbf{F}_D^3$, some unique high-level features apparently different from $\mathbf{M}_S$, $\mathbf{P}_S$, $\mathbf{F}_D^1$, and $\mathbf{F}_D^2$ are presented. All the various feature maps demonstrate that our method fully exploits the MS and PAN images' information.

## DISCUSSION

The proposed method has been compared with different pan-sharpening methods on reduced-resolution and full-resolution datasets. The parameter number and inference time of these methods have also been measured. It can be found that our method can yield superior results with minimum parameters and achieve a good trade-off between calculation and performance. The proposed method controls the number of parameters by keeping fewer feature maps throughout the network and turns to multi-resolution feature extraction for more effective feature maps. Visualizing these feature maps has verified that they have distinct content, which brings our method high efficiency.

By comparing the full model with the variants w/o MRFE and w/o SC, it can also be found that the HRFormer-like feature extractor is useful. The skip connections between every two resolutions have slightly improved the pan-sharpening performance.

As for the two-stage feature fusion, it can be observed that all indicators declined significantly, no matter removing the MRFM or the SDFM. Besides, the SDFM apparently boosts the convergence speed of our method. The variants w/o MRFE, w/o MRFM, w/o SC and the full model show the same convergence speed due to the SDFM. As the final fusion stage of diverse features, the SDFM inspired by residual learning significantly impacts the results and eases the gradient back-propagation. Among the variants, the variant w/o MRFE is simplest and can be regarded as a pure CNN-based baseline, which is easier to train than the transformer-based MRFE. Thus, the SDFM ensures the strength of the baseline, and the baseline guarantees the convergence speed of those variants with the MRFE.

## CONCLUSIONS

In this article, we proposed a pan-sharpening approach based on multi-resolution transformer and two-stage feature fusion. In the proposed network, two HRFormer-like structures formed a two-branch multi-resolution feature extractor to learn multi-scale and contextual feature maps from the MS and PAN images. A two-stage fusion scheme was proposed to fuse these diverse features. The MRFM fuses modality-specific multi-resolution features, and the SDFM finally fused shallow and deep features to generate the details to be injected into the MS image. Experiments on two kinds of datasets demonstrated the superiority of our method over state-of-the-art methods. The extracted feature maps were visualized to verify the diversity of the extracted features, and the ablation study also proved the effectiveness of the MRFE. Besides, the two-stage feature fusion is also proved to be

necessary *via* ablation experiments. The SDFM can even boost the convergence speed of the network. In future works, efforts will be made to enhance the time efficiency of transformer-based pan-sharpening methods.

### Funding

This work was supported by the National Natural Science Foundation of China (No. 61703299). The funders had no role in study design, data collection and analysis, decision to publish, or preparation of the manuscript.

### Grant Disclosures

The following grant information was disclosed by the authors:
National Natural Science Foundation of China: 61703299.

### Competing Interests

The authors declare there are no competing interests.

### Author Contributions

- Wensheng Fan conceived and designed the experiments, performed the experiments, analyzed the data, performed the computation work, prepared figures and/or tables, authored or reviewed drafts of the article, and approved the final draft.
- Fan Liu conceived and designed the experiments, authored or reviewed drafts of the article, and approved the final draft.
- Jingzhi Li performed the experiments, prepared figures and/or tables, and approved the final draft.

### Data Availability

The third-party QB raw data is available at GitHub: https://github.com/zhysora/PSGan-Family.

The third-party WV3 raw data is available at GitHub: https://github.com/zhysora/PGMAN.

The QB and WV3 datasets created from the raw data are available at Zenodo: Wensheng Fan. (2023). QB and WV3 Datasets for Pan-Sharpening [Data set]. Zenodo. Available at https://doi.org/10.5281/zenodo.7608683.

The codes are available at Zenodo: Vince Van. (2023). SomiFan/MRPT: Pan-Sharpening with Multi-Resolution Transformer and Two-Stage Feature Fusion (v1.0.1). Zenodo. Available at https://doi.org/10.5281/zenodo.7608885.

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
