# Peer review of "A pan-sharpening network using multi-resolution transformer and two-stage feature fusion"

_PeerJ Computer Science, doi:10.7717/peerj-cs.1488_

## Round 0.1 · original submission · Major Revisions

Revise as per reviewer comments.

Reviewer 1 ·

Basic reporting

The authors have provided background and sufficient literature references.

Experimental design

The authors did a lot of research and compared their method to the most recent techniques available from the literature.

Validity of the findings

'no comment'

Additional comments

The authors proposed a pan-sharpening network that utilizes multi-resolution transformers and two-stage feature fusion to enhance remote sensing images. Their method has excellent potential for practical applications. The authors conducted a thorough literature review and compared their approach with state-of-the-art techniques. The manuscript is well-structured. However, I have a few observations and suggestions regarding the manuscript:
1. In line no. 119, please correct “(80\%/10\%/10\%)”.
2. The manuscript’s English language could be improved to ensure that an international audience can understand it clearly. For instance, in some sentences, the current phrasing makes comprehension difficult (e.g., line no. 38: Due to physical constraints, optical remote sensing satellites can only provide MS images of a low spatial resolution but a high spectral resolution and panchromatic (PAN) images of a high spatial resolution but a low spectral resolution (Zhang, 2004; Zhou, Liu & Wang, 2021)”. I recommend that the authors ask a colleague proficient in English and familiar with the subject matter to review their manuscript or consider using a professional editing service.
3. Table-9 shows that DR-NET has more parameters than the proposed method, but its run time is less than the proposed method (0.0120 sec. vs. 0.0170 sec.), as is the case with MSDCNN. Therefore, what is the advantage of the proposed method’s low complexity?
4. Please mention the parameters’ units in Table-9.
5. In conclusion, please remove “we” from the last paragraph and rephrase it as “In future work, efforts will be made to enhance the time efficiency of transformer-based pan-sharpening methods.”
6. I commend the authors for their extensive research work and suggest that they incorporate the suggestions given to improve the manuscript’s quality before acceptance.”

·

Basic reporting

Excellent

Experimental design

Good and relevent

Validity of the findings

Appreciable

Additional comments

Review comments
Authors proposed a pan-sharpening scheme based on multi-resolution transformer and two-stage feature fusion in this paper. A transformer-based multi-resolution feature extractor is designed to extract multi-scale feature representations of the multi-spectral and panchromatic images. The paper is acceptable after addressing the following concerns. The review comments are:
1. Abstract looks little bit lengthy. Try to compress it.
2. More specific keywords can be used.
3. Revise the figure caption and table titles such that it should exactly reflect the content.
4. All variables and notions in the equation should be declared.
5. Now the text is a bit monotonous. All long sentences should split meaning fully split into short ones.
6. Future scope of the work should be included in the conclusion section.
So I requesting the editor to proceed with a minor revision.

Reviewer 3 ·

Basic reporting

This paper proposes a pan-sharpening method based on multi-resolution transformer and two-stage feature fusion. Overall this is a solid study. Some comments are as follows.
1. The research motivation is not convincing. There are already similar studies using Transformer in the literature. Their pros and cons should be summarized comprehensively.
2. The problem considered in this study is not fully described. In Materials & Methods-Datasets, the authors should add a problem description as well as the research challenges.
3. "80\%/10\%/10\%" should be "80%/10%/10%" in Line 119. Check Line 121 too.

Experimental design

1. The methods described in this study cannot be replicated with no dataset and code.

Validity of the findings

1. The data are not provided.

Additional comments

no comment

---

## Round 0.2 · accepted · Accept

The submission can be accepted now.

Reviewer 1 ·

Basic reporting

The authors have thoroughly and sincerely addressed all of the reviewer's comments. Therefore, I recommend the esteemed chief editor to accept this manuscript in its current form.

Experimental design

Good

Validity of the findings

Meaningful

·

Basic reporting

good

Experimental design

good

Validity of the findings

good

Additional comments

The authors have addressed all the comments that I suggested during the first review. The paper is acceptable in its present form.
So, I request the editor to ‘Accept’ the paper.

Reviewer 3 ·

Basic reporting

no comment

Experimental design

no comment

Validity of the findings

no comment